# Drip Irrigation Depth Alters Root Morphology and Architecture and Cold Resistance of Alfalfa

**Zhensong Li, Xianglin Li and Feng He \***

Institute of Animal Science, Chinese Academy of Agricultural Sciences, Beijing 100193, China
\* Correspondence: hefeng@caas.cn; Tel.: +86-10-6287-4535

**Abstract:** Combined stress from water and temperature is considered an effective approach for improving the cold resistance of alfalfa (*Medicago sativa* L.). However, the relationships among irrigation depth, root morphology and architecture, and cold resistance of alfalfa remain unclear. In this study, we investigated the effects of drip irrigation at the soil surface (Deep-0), at 20 cm depth (Deep-20), and at 40 cm depth (Deep-40) on root morphology and architecture and cold resistance of alfalfa. The Deep-0 treatment had the highest aboveground biomass and belowground biomass, and the root system in the Deep-40 treatment tended to a 'herringbone' branching type, which increases the cost of water uptake. After cold acclimation for 3 days, the cold resistance of Deep-0 was strongest, and was associated with higher soluble sugar content and fewer reactive oxygen species in the roots. Deep-40 enhanced the severity of drought stress on the crown and increased the risk of crown exposure to low-temperature stress. The results showed that Deep-0 promoted alfalfa growth and development by regulating root morphology and architecture and improving water absorption efficiency, thereby enhancing the ability of the root system to withstand low-temperature stress.

**Keywords:** alfalfa; drip irrigation depth; root morphological architecture; nonstructural carbohydrates; cold resistance





## 1. Introduction

Alfalfa (*Medicago sativa* L.) is a perennial leguminous herb that is cultivated worldwide owing to its excellent agronomic characteristics, such as yield, nutrition, and regenerability [1]. Alfalfa is mainly grown at high latitudes, which severely limits its regeneration capacity owing to the cold winters [2]. Optimized management practices are an important means by which to improve the overwintering of alfalfa, including sowing depth, sowing density, irrigation regime, fertilization, and grazing regime.

The root system of alfalfa is crucial for overwintering. The morphological and physiological characteristics of the roots are strongly associated with the cold resistance of alfalfa. An optimal root system architecture not only improves the efficiency of absorption of water and nutrients, but also enhances stress resistance [3]. Water is a crucial factor that affects the success of alfalfa overwintering because changes in soil moisture content have a greater impact on herbaceous plants [4,5]. On the one hand, sufficient water during the growth period promotes root development and storage of substances for energy; on the other hand, lower intracellular water content in winter reduces the freezing point [6], and freezing injury is mainly caused by cell dehydration [7,8]. Water not only affects the cold resistance of alfalfa by changing the root morphology and architecture, but also protects cells from low-temperature damage through physiological and metabolic pathways [9]. Plants can improve cold resistance by regulating root development in response to dynamic changes in soil moisture [10,11]. Belowground biomass [3,12], root crown [3,13], lateral roots [3,14], and the spatial distribution of roots [9] have been previously demonstrated to have important effects on the cold resistance of alfalfa. Optimal irrigation technology is a method to rapidly improve plant stress resistance. Previous studies have confirmed that

the combined stress of water and temperature can improve the cold resistance of alfalfa, which provides a scientific basis for field management of alfalfa in winter [7,15–17].

Compared with surface drip irrigation, subsurface drip irrigation is a more efficient and water-saving irrigation method. Subsurface drip irrigation has higher water use efficiency and results in higher yields [18]. Subsurface drip irrigation is generally considered to reduce evaporative water loss and promote growth of the root system, which has more obvious advantages than surface drip irrigation. However, a preliminary experiment showed that with increase in irrigation depth, the growth and development of alfalfa was inhibited, which was inconsistent with previous research results. To date, the few studies of subsurface drip irrigation have mainly focused on annual crop plants, such as wheat and corn; few studies have investigated perennial plants, and the effects on the root system and stress resistance of plants have not been considered. Therefore, by studying the effects of irrigation depth on the root morphology and architecture and cold resistance of alfalfa, the present experiment aimed to (1) clarify the response of root morphology and architecture to different irrigation depths, (2) explore the relationship between root traits and cold resistance, and (3) determine the influence of irrigation depth on the cold resistance of alfalfa.

## 2. Materials and Methods

### 2.1. Study Location and Plant Materials

This experiment was conducted in a controlled greenhouse at the Institute of Animal Science, Chinese Academy of Agricultural Sciences, Beijing, China, from May to September 2020. The greenhouse environment was 25 °C/20 °C (day/night), 14 h/10 h (light/dark) photoperiod, photosynthetic photon flux density of 350 mmol·m$^{-2}$·s$^{-1}$, and 60–65% relative humidity. Seeds of the alfalfa cultivar 'WL440′ (with a fall dormancy score of 6) were provided by the Beijing Zhengdao Seed Industry Co., Ltd. (Beijing, China). The seeds were sterilized with sodium hypochlorite (1% NaClO) for 30 min and washed with deionized water five times. Seeds of uniform size were germinated in a Petri dish under a 14 h/10 h (light/dark) photoperiod at 25 °C. After 72 h, three germinated seeds were transferred to one polyvinyl chloride (PVC) pipe with an inner diameter of 14 cm and height of 50 cm. A nylon mesh bag was placed in each pipe (to facilitate later sampling) and was filled with 2.5 kg sterilized dry sandy soil and nutrient soil mixture (1:4, *v/v*). The nutrient soil mixture is a cultivation medium (composed of peat moss and lime), named TS1, produced by Klasmann–Deilmann. TS1 contains 1.6% total nitrogen, 0.1% $P_2O_5$, 0.2% $K_2O$ (N:P:K = 14:10:18), and 91% organic matter, with conductivity 0.9 dS m$^{-1}$ and pH 5.8. The water-holding capacity (WHC) of the mixture was 38.35%. One plant was retained in each PVC pipe 2 weeks after transplanting according to its height (approximately 15 cm), and was cultivated for an additional 2 weeks before subsequent experimentation. The soil moisture content was maintained at 60–65% WHC by weighing each pipe every 4 days. Weeds and pests were removed regularly.

### 2.2. Experimental Design and Treatments

Three drip irrigation depths were applied in the experiment, namely, surface drip irrigation, subsurface drip irrigation at 20 cm depth, and subsurface drip irrigation at 40 cm depth (designated Deep-0, Deep-20, and Deep-40, respectively). We used a PVC hose with an inner diameter of 3 mm to transport the water to the specified depth. Each treatment comprised 10 replicates. The soil moisture content was maintained at 60–65% WHC by weighing each pipe of Deep-0 every 4 days, and the irrigation amount of Deep-20 and Deep-40 was the same as Deep-0. After 8 weeks we conducted phase 1 sampling (Figure 1), with five replicates sampled per treatment. The aboveground and belowground parts of the seedlings were separated. The roots were carefully removed from the nylon mesh bag in each pipe to minimize damage to the spatial distribution of the root system. The root surface was gently washed manually with distilled water and the root system was arranged evenly in a transparent acrylic tray containing 1200 mL distilled water. The root system was

scanned with a MICROTEK Scan Maker i800 plus (Microtek Technology Co., Ltd., Shanghai, China) with a resolution of 600 dpi. Immediately after scanning, approximately 5 cm of the root crown was used to determine the electrical conductivity and physiological indicators. We divided this sample into two portions: one portion was used for determination of the semi-lethal temperature ($LT_{50}$), and the other portion was used for determination of physiological indicators after storage at $-80$ °C. The remainder of the root system was used to calculate the biomass. The aboveground and underground parts were weighed after oven drying at 65 °C for 48 h. The dry weights were regarded as the aboveground biomass (AGB) and belowground biomass (BGB).

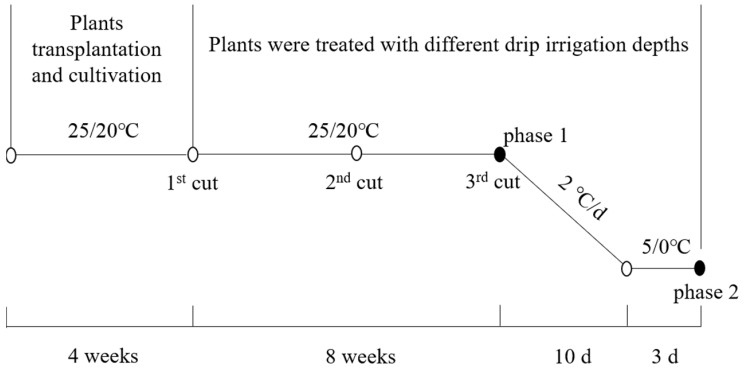

**Figure 1.** Schematic diagram of the experimental and sampling process.

The remaining half of the experimental plants were moved to an LRH-200-GD low-temperature light incubator (Taihong Medical Instruments, Guangzhou, China) for the low-temperature treatment (phase 2). The initial temperature was 25 °C/20 °C (day/night) with a photoperiod of 10 h/14 h (light/dark), and the photosynthetic photon flux density was 350 mmol·m$^{-2}$·s$^{-1}$. The temperature was decreased to 5 °C/0 °C (day/night) at the rate of 2 °C·d$^{-1}$ and the light intensity was decreased to 150 mmol·m$^{-2}$·s$^{-1}$ at the rate of 20 mmol·m$^{-2}$·s$^{-1}$·d$^{-1}$, simulating the cold-adaptation environment of alfalfa. Sampling was conducted after a further 72 h of cold acclimation. During this phase of the experiment, plants were watered as in phase 1. The roots were carefully rinsed by hand with distilled water after the test and the root crown sample was used for the measurement of electrical conductivity.

### 2.3. Root Morphology and Architecture

We used Win-RHIZO 2017a (Regent Instruments, Inc., Quebec, QC, Canada) to analyze the scanned images (Figure 2). The process included thresholding, framing, editing breakpoints, and eliminating loops to determine the following root morphological indicators: root length, root surface area (RSA), root volume (RV), root forks (RF), and average link length (ALL). Specific root length (SRL) was calculated by total root length/BGB.

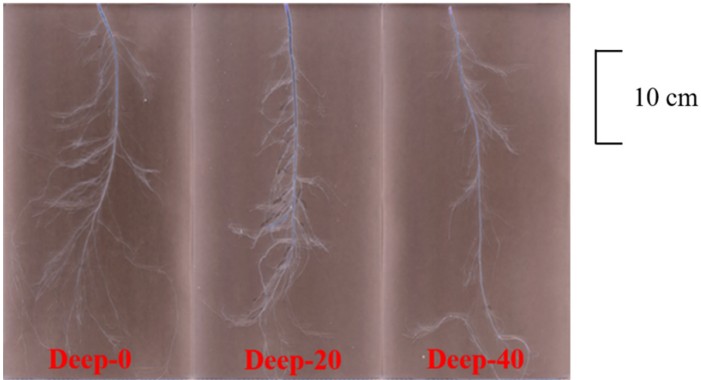

**Figure 2.** Scanned images of the alfalfa root system under three drip irrigation depths.

The topological index (TI) is used as a measure of the spatial structure of a root system and is defined as log altitude ($A$)/log magnitude ($M$), where $A$ is the number of links in the longest path from an exterior link to the most basal link of the root system and $M$ is the total number of exterior links. When TI is close to 0.5, the root system tends to dichotomous branching and when close to 1 it tends to herringbone branching [19]. The fractal dimension (FD) was calculated using the box-dimension method [20].

### 2.4. Semi-Lethal Temperature

The $LT_{50}$, the temperature at which the relative permeability of intracellular ions attains 50%, was used to represent the cold resistance of alfalfa in this study [15,21]. We considered the 5 cm underground portion of the taproot as the root crown, which was cut into nine pieces of 2–3 mm length. Each piece was placed into a 2 mL centrifuge tube. The tubes were incubated at 8 °C for 2 h. The subsequent freezing test was conducted in a ZX-5C constant-temperature circulator (Zhixin Instrument, Shanghai, China) under a decreasing series of nine temperatures. The samples were incubated in alcohol in the tubes for 1.5 h at each temperature. The alcohol temperature differed because of the difference in $LT_{50}$ under the two treatments. For the samples collected in phase 1, the nine temperatures were set to 8, 6, 4, 2, 0, −2, −4, −6, and −8 °C. For the samples collected in phase 2, the nine temperatures were set to 0, −2, −4, −6, −8, −10, −12, −14, and −16 °C. After 1.5 h at the first temperature in each phase, one tube was transferred to storage at that temperature; after 1.5 h at the second temperature, a second tube was removed for storage at that temperature, and so forth, until all nine tubes in each phase were stored at their designated temperatures. We then transferred the pieces of root crown from the 2 mL tube to a 15 mL tube and added 5 mL deionized water. This tube was shaken on an HZQ-A gyratory platform shaker (Hengrui Instrument and Equipment, Changzhou, China) at 120 rpm for 12 h at 25 °C. Using a FE38 conductivity meter (Mettler, Shanghai, China), the electrical conductivity was measured (designated $EL_1$). The sample was autoclaved at 121 °C for 30 min and, on remeasuring, its electrical conductivity was designated $EL_2$. The electrical conductivity of deionized water was designated EL. Relative electrolyte leakage was calculated according to Equation (1) and the semi-lethal temperature was calculated with Equation (2):

$$\text{Relative electrolyte leakage (\%)} = (EL_1 - EL)/(EL_2 - EL) \times 100 \tag{1}$$

$$y = A/(1 + B \times e^{-kx}) \times 100 \tag{2}$$

where $x$ is the freezing temperature, $y$ is the relative electrical leakage, and $A$, $B$, and $k$ are constants.

### 2.5. Root Physiological Indicators

Physiological indicators were measured using the root crown sampled in phase 1. The portion of the crown sample stored at −80 °C was ground into powder and used to determine the malondialdehyde (MDA), superoxide anion ($O^{2-}$), hydrogen peroxide ($H_2O_2$), and proline contents, peroxidase (POD), catalase (CAT), and superoxide dismutase (SOD) activities, and soluble sugar, sucrose, fructose, glucose, trehalose, and starch contents. All physiological indicators were determined in accordance with the manufacturer's instructions for the corresponding commercial kits (Beijing Box Bio-engineering Technology Co., Ltd., Beijing, China).

Malondialdehyde reacts with thiobarbituric acid to produce a reddish-brown substance with maximum absorbance at 532 nm. Therefore, the MDA content was calculated according to the absorbance value. Superoxide anion reacts with hydroxylamine to form nitrite, which produces a pink substance under the action of 4-aminobenzenesulfonic acid and $\alpha$-naphthylamine, with maximum absorbance at 530 nm. Thus, the activity of $O^{2-}$ was calculated according to the absorbance value.

In an acidic solution, $H_2O_2$ and potassium permanganate react to make the solution a reddish color. The content of $H_2O_2$ was calculated based on the volume of potassium permanganate. Proline is red after heating with acidic ninhydrin. The maximum absorbance is measured at 520 nm. The content of proline was calculated according to the absorbance value. Peroxidase oxidizes guaiacol to produce a tawny-brown substance. The activity of POD was calculated according to the absorbance at 470 nm. Catalase decomposes hydrogen peroxide and the absorbance of the reaction solution at 240 nm decreases with the duration of the reaction. The activity of CAT was calculated according to the change in absorbance. Under light conditions, $O^{2-}$ can reduce nitro-blue tetrazolium to the blue methyl hydrazone, for which the maximum absorbance is measured at 560 nm. Superoxide dismutase scavenges $O^{2-}$, thereby inhibiting the photoreduction reaction of nitro-blue tetrazolium and reducing the rate of formation of blue methyl hydrazone. Therefore, SOD activity was calculated from the absorbance at 560 nm.

Soluble sugars react with concentrated sulfuric acid and anthrone to generate blue-green derivatives that have maximum absorbance at 620 nm. The soluble sugar content was calculated based on the absorbance. Sucrose is converted into reducing sugar by hydrochloric acid hydrolysis. The difference in amount of reducing sugar before and after the reaction was determined as the sucrose content. Fructose reacts with resorcinol to form a colored substance under an acidic condition and the product has a characteristic absorption peak at 480 nm. The content of fructose was determined from the change in absorbance. Glucose oxidase oxidizes glucose to gluconic acid and releases $H_2O_2$, which condenses with a chromogenic oxygen acceptor under the catalysis of peroxidase to form a red compound. The maximum absorption peak is measured at 505 mm and the absorbance value is proportional to the amount of glucose. Trehalose is dehydrated under concentrated acidic conditions to produce 5-hydroxymethylfurfural, which reacts with anthrone to generate a blue-green furfural derivative. The product has maximum absorbance at 620 nm. The trehalose content was calculated from the absorbance value.

### 2.6. Statistical Analyses

The Shapiro–Wilk test and Levene test showed that all data in this experiment conformed with a normal distribution and satisfied the homogeneity of variance. The data were subjected to analysis of variance between treatments using IBM SPSS Statistics 20.0 (IBM Corporation, Armonk, NY, USA) followed by a least significant difference (LSD) test to determine the means that differed significantly. Differences were considered significant at the 5 or 1% significance levels.

## 3. Results
### 3.1. Biomass

The differences in growth of alfalfa plants under the different irrigation treatments are illustrated in Figure 3. In all treatments, the AGB was highest in the first cut, followed by the second cut, and lowest in the third cut (Figure 4a). Significant differences in AGB of the three cuts were observed among the three irrigation depths ($p < 0.05$). The AGB in the first cut of Deep-0 and Deep-20 was 2.26 and 1.96 g·plant$^{-1}$, respectively, which was significantly higher than that of Deep-40. In the second cut, the AGB of Deep-0 and Deep-20 was not significantly different, but was significantly higher than that of Deep-40. The AGB of Deep-0 was the largest in the third cut, which was significantly higher than that of Deep-20 and Deep-40. Deep-40 had the lowest BGB (1.16 g·plant$^{-1}$), which was significantly lower than that of Deep-0 and Deep-20. Deep-40 had the highest root:shoot ratio (1.31) among the three irrigation treatments, which was significantly higher than that of Deep-20 (1.08) and Deep-0 (0.81).

### 3.2. Root Morphological Indicators

Significant differences in specific root length (SRL), primary root length (PRL), total lateral root length (TLRL), crown diameter (CD), RSA, and RV among the three irrigation

depths were observed ($p < 0.05$, Figure 5). The PRL of Deep-40 was 50.20 cm, which was significantly longer than that of Deep-0 (43.46 cm) and Deep-20 (40.62 cm). The TLRL decreased with the increase in irrigation depth. Thus, the TLRL of Deep-0 was the longest at 313 cm, which was not significantly different from that of Deep-20 but significantly longer than that of Deep-40. The CDs of Deep-0 and Deep-20 were 3.55 and 3.45 mm, respectively, which were significantly broader than that of Deep-40. The RSA and RV of Deep-0 ($81.2 \text{ cm}^2$ and $1.22 \text{ cm}^3$, respectively) were highest among the three irrigation depths.

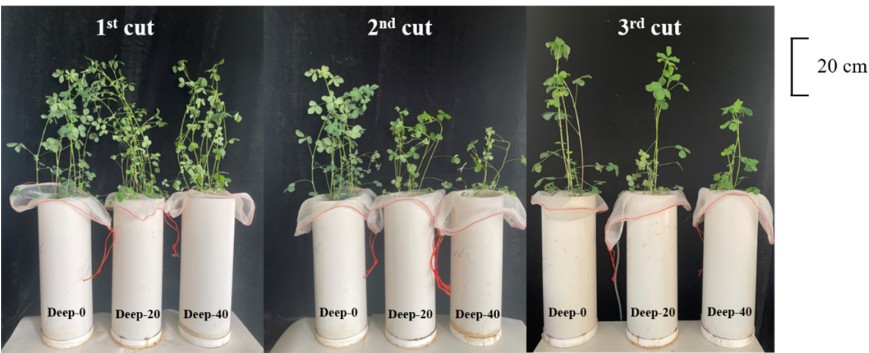

**Figure 3.** Growth of alfalfa plants under three drip irrigation depths.

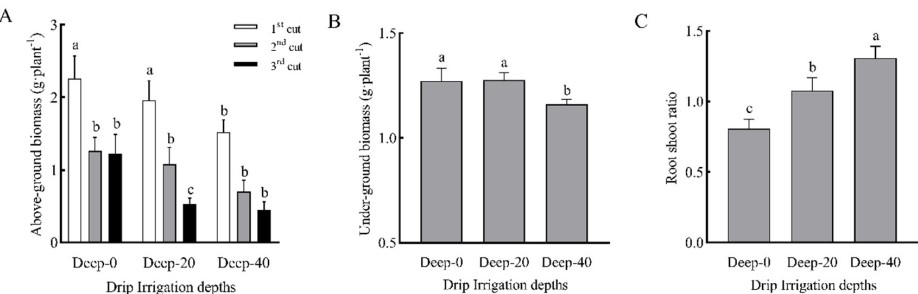

**Figure 4.** Biomass allocation of alfalfa plants under three drip irrigation depths. (**A**) Aboveground biomass, (**B**) belowground biomass, and (**C**) root:shoot ratio. Lower-case letters above bars indicate a significant difference ($p < 0.05$, LSD test).

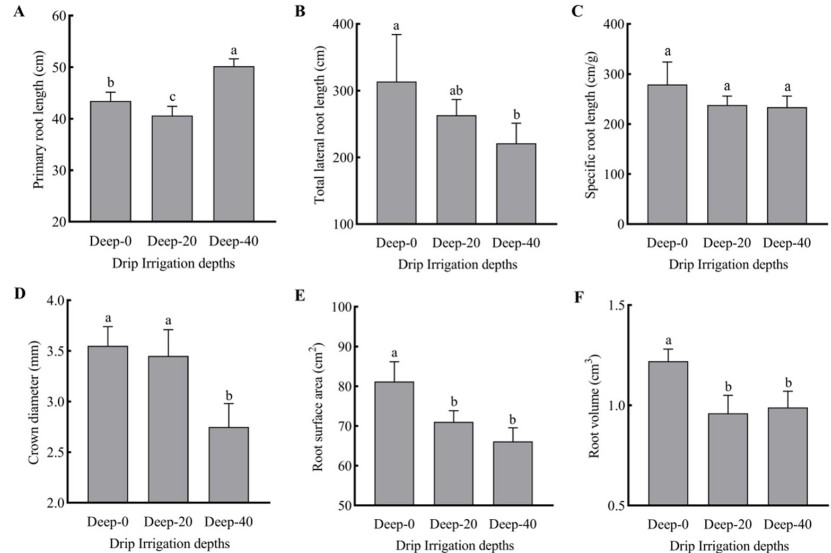

**Figure 5.** Root morphological traits of alfalfa plants under three drip irrigation depths. (**A**) Primary root length, (**B**) total lateral root length, (**C**) specific root length, (**D**) crown diameter, (**E**) root surface area, and (**F**) root volume. Lower-case letters above bars indicate a significant difference ($p < 0.05$, LSD test).

### 3.3. Root System Architecture

No significant effect on ALL, FD, and branch angle of the root system was observed among the three irrigation treatments ($p > 0.05$), although a trend for increases in ALL and FD with the increase in irrigation depth was evident (Figure 6). These results indicated that there was a slight increase in soil spatial occupancy of the root system. Irrigation depth had a significant effect on RF ($p < 0.01$). The number of root forks of Deep-0 (879) was significantly higher than that of the other two treatments. The TI differed significantly among the three treatments ($p < 0.01$). With increase in irrigation depth, the TI of the root system increased and the root architecture tended to assume a 'herringbone' branching type.

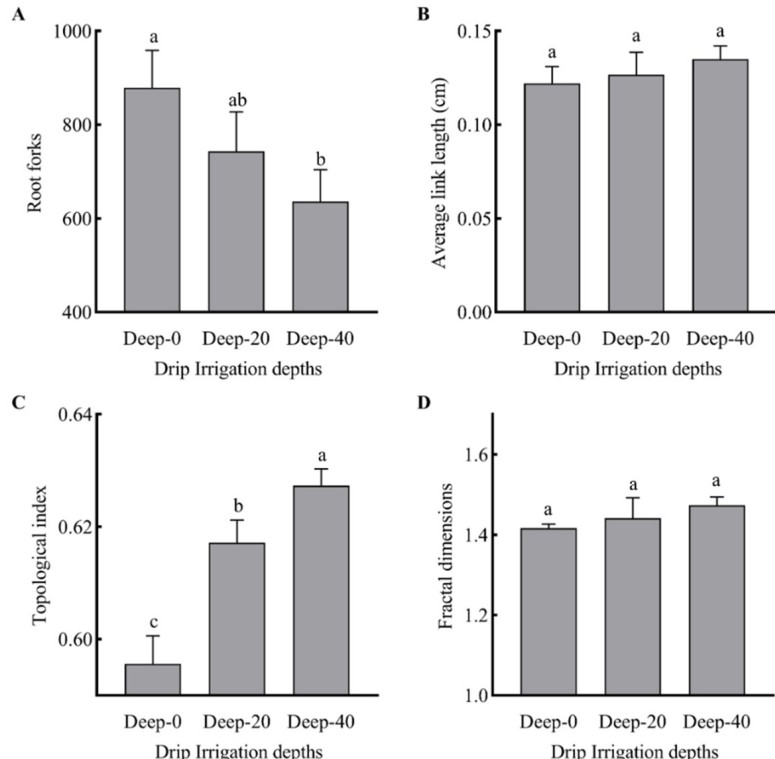

**Figure 6.** Root system architecture of alfalfa plants under three drip irrigation depths. (**A**) Number of root forks, (**B**) average link length, (**C**) topological index, and (**D**) fractal dimension. Lower-case letters above bars indicate a significant difference ($p < 0.05$, LSD test).

### 3.4. Semi-Lethal Temperature

The changes in $LT_{50}$ of alfalfa crowns at phases 1 and 2 among the three treatments are shown in Figure 7. After cold acclimation (phase 2), the $LT_{50}$ of the three treatments showed a strongly significant decrease ($p < 0.01$). The decrease in $LT_{50}$ of Deep-0 was largest, attaining 7.40 °C, whereas that of Deep-20 and Deep-40 decreased by 6.82 and 5.28 °C, respectively. No significant difference in $LT_{50}$ among the different irrigation depths was detected in phase 1; the $LT_{50}$ of the three irrigation depths at this stage were 1.88, 2.48, and 2.46 °C, respectively. At phase 2, the $LT_{50}$ showed a trend to increase with the greater depth of irrigation. The $LT_{50}$ of Deep-40 ($-2.82$ °C) was significantly higher than that of Deep-20 ($-4.34$ °C) and Deep-0 ($-5.52$ °C).

### 3.5. Correlation Analysis between $LT_{50}$ and Phenotypic Traits

Significant correlations of AGB and BGB with root morphological traits were observed (Figure 8). The increases in TLRL, CD, RSA, RV, and RF were beneficial for improved biomass, especially for AGB. The root:shoot ratio was significantly negatively correlated with TLRL. The TLRL was significantly positively correlated with RSA and RV, and signifi-

cantly negatively correlated with TI. The RSA was significantly positively correlated with RV and RF, and significantly negatively correlated with TI. No significant correlation was observed between root traits and $LT_{50}$ at phase 1, but significant correlations were observed at phase 2. The increases in AGB, BGB, TLRL, CD, RSA, RV, and RF were beneficial for improved cold resistance, whereas increases in PRL, TI, and FD were detrimental to the cold resistance of alfalfa.

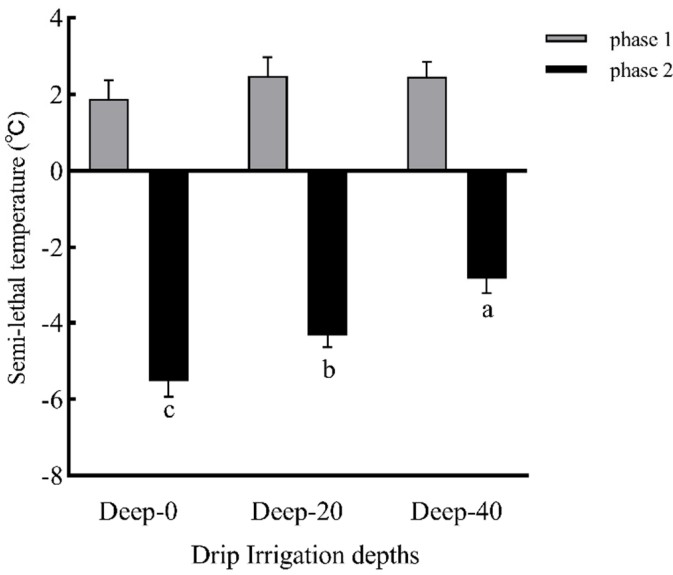

**Figure 7.** Semi-lethal temperature of alfalfa crowns under three drip irrigation depths at the phase 1 and 2 sampling time points. Lower-case letters above bars indicate a significant difference ($p < 0.05$, LSD test).

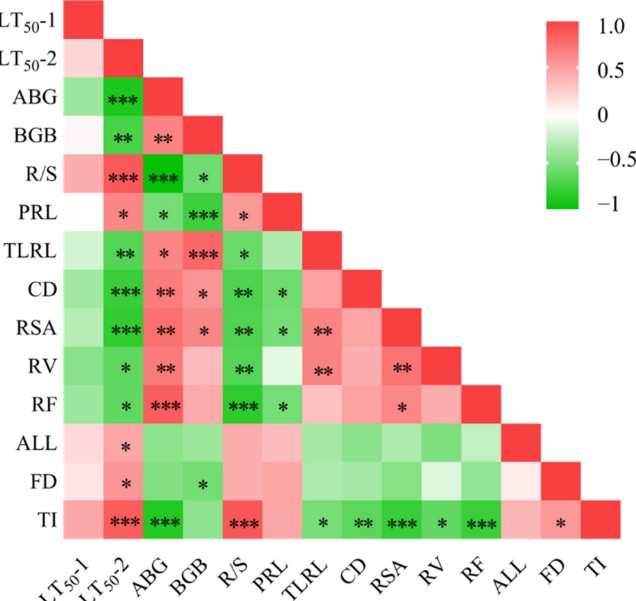

**Figure 8.** Pearson correlation analysis of semi-lethal temperature ($LT_{50}$) and phenotypic traits of alfalfa under three drip irrigation depths. $LT_{50}{}^{-1}$, semi-lethal temperature of phase 1; $LT_{50}{}^{-2}$, semi-lethal temperature of phase 2; AGB, aboveground biomass; BGB, belowground biomass; R/S, AGB/BGB; PRL, primary root length; TLRL, total lateral root length; CD, crown diameter; RSA, root surface area; RV, root volume; RF, number of root forks; ALL, average link length; TI, topological index; FD, fractal dimension. *, ** and *** represent a significant correlation at the 0.05, 0.01 and 0.001 levels, respectively.

### 3.6. Reactive Oxygen Species and MDA

Irrigation depth significantly affected ($p < 0.05$) the MDA content and $O^{2-}$ activity, but had no significant effect ($p > 0.05$) on $H_2O_2$ content (Figure 9). The MDA content of Deep-0 was the lowest (111.84 nmol/g FW), which was significantly lower than that of Deep-20 and Deep-40. Activity of $O^{2-}$ in the Deep-40 treatment (171.39 U/g FW) was significantly higher than that of Deep-20 and Deep-0. The $H_2O_2$ content of Deep-40 was the highest (46.93 µmol/g FW), followed by Deep-20 and Deep-0 (41.00 and 36.22 µmol/g FW, respectively).

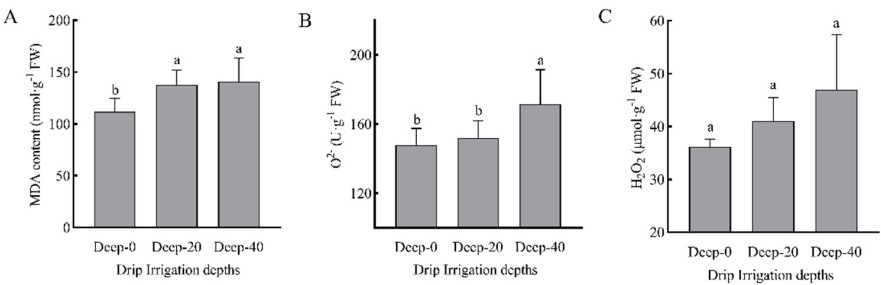

**Figure 9.** Malonaldehyde and reactive oxygen species contents in the crown of alfalfa plants under three drip irrigation depths. (**A**) Malonaldehyde, (**B**) superoxide anion ($O^{2-}$), and (**C**) hydrogen peroxide ($H_2O_2$). Different lower-case letters above bars indicate a significant difference between irrigation depths ($p < 0.05$, LSD test).

### 3.7. Antioxidant System Activity

The proline content of Deep-0 (8.85 mg/g FW) was lower than that of Deep-20 and Deep-40 (10.46 and 10.95 mg/g FW, respectively), but the differences among the three irrigation depths were not significant ($p > 0.05$, Figure 10). The POD activity was elevated with increase in depth of irrigation. The POD activity of Deep-0 (129.46 U/g FW) was significantly lower ($p < 0.05$) than that of Deep-20 and Deep-0. The CAT activity of Deep-40 (147.48 U/g FW) was significantly higher ($p < 0.05$) than that of Deep-0 (114.57 U/g FW), but not significantly ($p > 0.05$) different from that of Deep-20 (128.30 U/g FW). The SOD activity of Deep-40 (145.80 U/g FW) was significantly higher than that of Deep-0 (111.55 U/g FW).

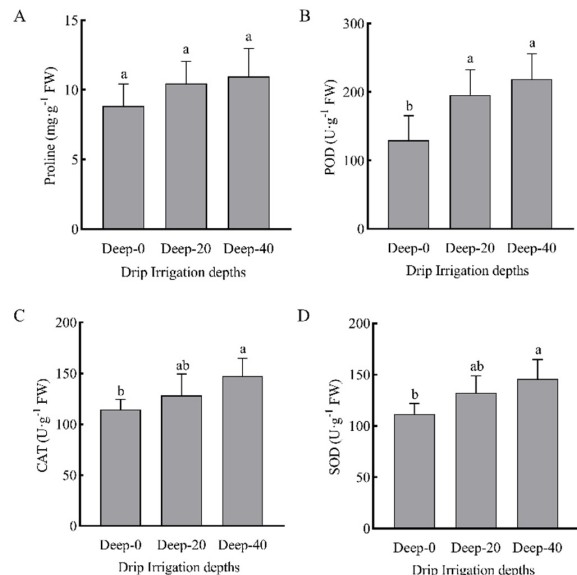

**Figure 10.** Proline content and antioxidant enzyme activities in the crown of alfalfa plants under three drip irrigation depths. (**A**) Proline, (**B**) peroxidase (POD), (**C**) catalase (CAT), and (**D**) superoxide dismutase (SOD). Different lower-case letters above bars indicate a significant difference between irrigation depths ($p < 0.05$, LSD test).

### 3.8. Non-Structural Carbohydrates

The soluble sugar content of Deep-0 (54.53 mg/g FW) was significantly higher than that of Deep-20 (45.22 mg/g FW) and Deep-40 (43.97 mg/g FW) (Figure 11). Excluding trehalose, the contents of sucrose, glucose, and fructose were consistent among the three treatments; the contents in the Deep-0 treatment (28.71, 15.31, and 13.21 mg/g FW, respectively) were significantly higher ($p < 0.05$) than those of the other two treatments.

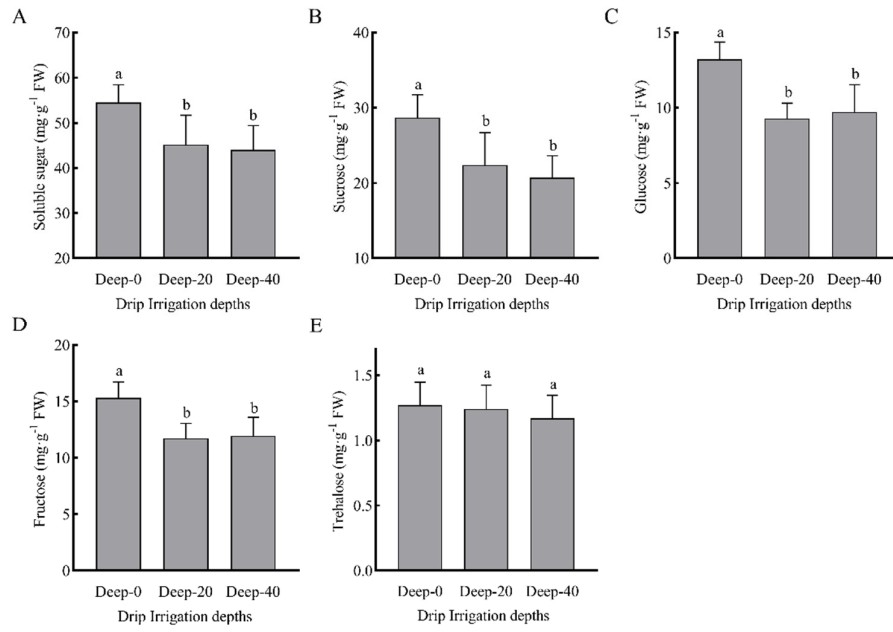

**Figure 11.** Non-structural carbohydrate contents in the crown of alfalfa plants under three drip irrigation depths. (**A**) Soluble sugar, (**B**) sucrose, (**C**) glucose, (**D**) fructose, and (**E**) trehalose. Different lower-case letters above bars indicate a significant difference between irrigation depths ($p < 0.05$, LSD test).

## 4. Discussion

Irrigation depth is an important factor affecting root growth, especially for herbaceous plants [22,23]. Research on wheat has shown that the deeper the irrigation depth, the greater the root biomass in the deep soil. Improvement of the water use efficiency of roots and reduction in surface water evaporation can help increase yields and conserve water [24]. However, the present results showed that the aboveground biomass of alfalfa decreased consistent with increase in irrigation depth, which may reflect climatic differences at different experimental sites. In addition, the lack of supplemental irrigation might result in low surface water content, which may affect plant growth and development. Shallow roots were more frequent when the surface soil was wet, whereas deep roots only developed when the surface soil was subjected to water stress. Root plasticity is an important strategy for plants to cope with soil heterogeneity [25,26], in which lateral roots are the main organs for absorption of water and nutrients. The present study showed that the lateral root length and surface area in the Deep-0 treatment were significantly higher than in the other treatments, which accounted for the higher AGB and BGB of the Deep-0 treatment [10,27,28]. From an energy perspective, the absorption of water from deep soil by roots increased the energy consumption of water transport, which was not conducive to the plant coping with stress [28–30]. Although a greater number of lateral roots and root surface area increased the redundant consumption, the overcompensation effect of root biomass exceeded the redundant consumption, hence the root growth under surface drip irrigation was superior [31].

The higher RF of the Deep-0 treatment was indicative of greater water uptake efficiency of alfalfa roots [32–34]. The root architecture under the Deep-40 treatment tended to a 'herringbone' branching type, which was consistent with previous results under a low-

frequency irrigation treatment [10,35,36], both of which are associated with shallow-soil water deficit. The strong plasticity of plant roots in different soils stems from their avoidance of adversity. Even in soils with the same water content, uneven water distribution will cause drought stress to roots of the dry soil layer, and the root architecture will change from the original 'dichotomous' branching type to a 'herringbone' branching type [37]. Water deficit may inhibit the formation of lateral roots, thereby changing the morphological characteristics of roots. A study of Arabidopsis found that, under moderate drought stress, the primary root was better developed and the lateral roots were less developed; however, with increase in the duration of drought stress, growth of the primary root was also inhibited [38]. In soil with an uneven water distribution, water tropism determines that the root system will allocate more resources to the portions where growth is vigorous, to enable the root system to absorb greater quantities of water to support growth of the plant. The different depths of drip irrigation would have resulted in differences in the characteristics of soil moisture volume. Drip irrigation at different depths can regulate the growth and distribution of roots [39]. The root system of alfalfa is mainly distributed in the topsoil and the proportion of roots decreases with increase in depth. However, increasing the depth of drip irrigation increased the proportion of deep roots, including root biomass, root length, root surface area, and root volume [40].

Under water deficit, the balance between intracellular free radicals, reactive oxygen species, and reactive oxygen scavengers will be disrupted. With the increase in water stress, the removal of intracellular reactive oxygen species slows down, and excessive reactive oxygen species can inhibit the metabolic activity of cells [41]. The present study showed that the root crown of alfalfa was gradually subjected to drought stress with increase in irrigation depth, resulting in increased MDA content and $O^{2-}$ activity; in addition, antioxidant enzyme activities in the roots continued to increase, which played an important role in removing excess MDA and $O^{2-}$, and thus maintaining redox homeostasis [42–44]. Carbohydrates not only act as osmotic regulators to protect the integrity of cell structure and function, but are also intracellular signaling molecules that regulate gene expression during plant growth and development, and are strongly associated with plant cold resistance [45,46]. Increase in the contents of total soluble sugars, sucrose, glucose, fructose, and trehalose in cells is beneficial to improve the cold resistance of plants [8,45,47,48]. In the current study, surface drip irrigation (Deep-0) was more conducive to accumulation of soluble sugars, sucrose, glucose, and fructose in the root crown, whereas trehalose content showed no significant difference among the treatments, which was consistent with the trend in change of $LT_{50}$.

The root system is crucial to the overwintering of alfalfa, but the relationship between root morphological traits and cold resistance remains unclear [3]. Previous studies of alfalfa have mainly focused on the root crown; comparison of cultivars differing in cold resistance has revealed that the root crown depth of cultivars with strong cold resistance is generally deeper, which is considered to be a cold-sensitivity escape mechanism to prevent exposure of the overwintering tissue to low temperature [14]. Excluding the root crown, other root traits were also significantly correlated with $LT_{50}$, which was mainly shown after cold acclimation. Thus, BGB, TLRL, CD, RSA, RV, and RF were significantly negatively correlated with $LT_{50}$, whereas PRL, TI, and FD were significantly positively correlated with $LT_{50}$. These results seem to contradict the aforementioned conclusion that the difference in cold resistance among cultivars mainly reflects differences in development of the root system. Given that overwintering of alfalfa is an energy-consuming process, a large difference in biomass will mask the differences caused by other root system traits, such as TLRL and RSA [49]. In combination with previous research, the present findings show that water has a regulatory effect on root traits, and that root traits are strongly associated with the cold resistance of alfalfa plants. Therefore, it is feasible to rapidly improve the cold resistance of alfalfa through optimization of the irrigation regime. However, given the current paucity of relevant studies, the present results need further verification [3,7,15].

## 5. Conclusions

The present study has shown that drip irrigation depth has a strong influence on root morphology and architecture and cold resistance of alfalfa. Surface drip irrigation promoted plant growth and development, including increasing aboveground biomass, belowground biomass, lateral root length, root crown diameter, root surface area, root volume, and number of root forks, which are all beneficial traits to improve root water absorption efficiency and nutrient uptake. After exposure to low-temperature stress (phase 2), the $LT_{50}$ was negatively correlated with root biomass and volume. Drip irrigation at 40 cm depth promoted accumulation of reactive oxygen species in the root crown and decreased accumulation of non-structural carbohydrates, thereby increasing the $LT_{50}$ of alfalfa. These results suggest that surface drip irrigation improves the water use efficiency and enhances the cold resistance of alfalfa.

**Supplementary Materials:** The following supporting information can be downloaded at: https://www.mdpi.com/article/10.3390/agronomy12092192/s1, Table S1: Raw Data Table.

**Author Contributions:** Z.L., F.H. and X.L. conceived and designed the experiments; Z.L. performed the experiments, analyzed the data, prepared figures and tables, wrote and reviewed drafts of the paper, and approved the final draft; F.H. and X.L. reviewed drafts of the paper, and approved the final draft. All authors have read and agreed to the published version of the manuscript.

**Funding:** This research was supported by the National Natural Science Foundation of China (No.32071880), the Agricultural Science and Technology Innovation Program (ASTIP-IAS14) and the earmarked fund for CARS (CARS-34).

**Institutional Review Board Statement:** Not applicable.

**Informed Consent Statement:** Not applicable.

**Data Availability Statement:** The data presented in this study are available in (Supplementary Materials Table S1).

**Conflicts of Interest:** The authors declare no conflict of interest.

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
