# Peer review of "Drip Irrigation Depth Alters Root Morphology and Architecture and Cold Resistance of Alfalfa"

_agronomy, doi:10.3390/agronomy12092192_

Round 1

Reviewer 1 Report

This paper attempts to describe some root and shoot morphological and physiological responses of an alfalfa cultivar to irrigation depth. Such information will contribute to the body of knowledge about the role of irrigation depth in the response of alfalfa to abiotic and biotic stress. In this particular study, the focus is on cold response. The current manuscript can be substantially shortened without affecting the content. 

A main concern is it appears that there was no duplicate experiment performed? It is routine for greenhouse studies to be repeated.

General comments:

Water uptake efficiency – the authors suggested that larger root surface area and lateral root length improved “water uptake efficiency”. There appears to be no direct measurement of this trait so such a claim cannot be made given the data.

Growth conditions – what was the basis for the 14h/10h (light/dark) photoperiod? Is this photoperiod representative of conditions leading up to cold stress? Would the response vary if the light phase was shorter? Please indicate planting date or dates where applicable.

Methods:

Page 3, L46. What is meant by “Root Morphology Architecture”? In in the rest of the manuscript “morphology” and “architecture” are treated separately. It might be desirable to clarify?

Page 3, L47-52. Specific root length is not defined (this measurement is shown in Fig. 5)

Page 4, Figure 2. Please include a scale. It would be helpful to label the main root and first or second order lateral roots, where applicable

Page 4, L24-25. Please cite reference for the box-dimension method.

Please cite a reference for “herringbone” branching type if a reference is available.

Page 6, Figure 3 – please include a scale

Page 7, Figure 5. Plate C lacks significance letters

Page 8, Figure 6. Plates B and D lack significance letters

Page 10, Figure 9. Plate C lacks significance letters

Page 10, Figure 10, Plate A lacks significance letters

Page 11, Figure 11. Plate E lacks significance letter

Page 11, L49-50. Please explain differences of response at different experimental sites? Which experimental sites are referenced?

Reviewer 2 Report

Well designed and well written!

Author Response

It is a great honor to receive your recognition for this manuscript.

Reviewer 3 Report

The main goals of the study were to investigate the effects of drip irrigation at the soil surface, at 20 cm depth, and at 40 cm depth on root morphology and architecture and cold resistance of alfalfa. The Authors carried out the preliminary experiment in a controlled greenhouse conditions in 2020. The Authors showed  interesting results. They showed that surface drip irrigation  promoted alfalfa growth, development by regulating root morphology, architecture, improving water absorption efficiency, what enhancing the ability of the root system to withstand low temperature stress.  Work written correctly,  research methods selected correctly as well, results presented in a clear manner, enough literature included, conclusions answering for main goals of the work. There is one question pointed in the text about irrigation. Please correct or explain.
